# Multifactorial Study on the Impact of Educational Level, Employment Status, and the Need for Extraordinary Care on the Economic Impact of Cancer Patients

**DOI:** 10.3390/healthcare11091306

**Published:** 2023-05-03

**Authors:** Alberto García Martín, Eduardo J. Fernández Rodríguez, Celia Sánchez Gómez, M. Isabel Rihuete Galve

**Affiliations:** 1Department of Labour Law and Social Work, University of Salamanca, 37007 Salamanca, Spain; 2Department of Nursing and Physiotherapy, University of Salamanca, 37007 Salamanca, Spain; 3Institute of Biomedical Research of Salamanca (IBSAL), 37007 Salamanca, Spain; 4Department of Developmental and Educational Psychology, University of Salamanca, 37007 Salamanca, Spain; 5Medical Oncology Unit, University Hospital of Salamanca, 37007 Salamanca, Spain

**Keywords:** access to care, cancer, inequality, employment situation, healthcare utilization, disparity, oncology

## Abstract

Cancer is one of the major socio-health problems in the world. The level of education, the profession and/or employment status of the patient and the family can influence the amount of household income, the additional expenditure, and the possible socio-economic impact of the disease. The main objective of the study is to analyse and evaluate the influence of the level of education, the employment status of the patient, and the need for care and how these factors influence the additional expenditure and the possible socio-economic impact. Methods: descriptive cross-sectional randomised observational epidemiological study without replacement at the Hospital Universitario de Salamanca (CAUSA). Results: total sample (n = 365) comprised 53.2% of patients with no education or primary education, 25.8% with secondary education and 21.1% with higher education. Overall, 36.4% of patients were employed, 10.1% were self-employed, 53.0% were not employed, and 38.9% were experiencing other conditions. Significant statistics were found for educational level, employment status of the patient and main caregivers in terms of financial expenditure. Conclusions: Oncology patients with more education spend more on home help and/or accompanying the patient and transfers to the hospital for treatment. Higher incomes are not synonymous with higher expenditure in the sample. The patient’s main caregivers are a fundamental pillar of the patient’s household income and additional spending on orthopaedic material.

## 1. Introduction

Cancer is the second leading cause of death in the world [1], the first cause of death in men and the second in women [2,3]. The disease poses a major challenge to society as a whole due to its high incidence, mortality, and prevalence [4,5,6,7].

The diagnosis of cancer has a major impact on all areas of the life of the person diagnosed and also on the family and their closest direct environment; it affects patients and families on psychological, social, occupational, family, emotional, etc., levels. [6,7,8,9,10]. From the moment of oncological diagnosis, the patient faces a number of biological, psychological, and social changes, having to solve or resolve complex problems related to physical mobility, autonomy, work situation, family relationships, personal relationships, capacity for self-care, reduction of income, need for adaptation of the environment, and acquisition of orthopaedic material, among others [11,12,13,14]. The employment situation at the time when the patient receives the diagnosis can be a cause or aggravating factor in the socio-economic situation for the subsistence of the patient and the family, determining the evolution of the patient’s own oncological disease [15,16,17]. It is difficult to separate the employment situation from the patient’s level of academic studies; depending on the conditions of the latter, patients may have better jobs that may or may not make active treatment compatible with being able to continue working.

The public benefits of the health care system are insufficient for cancer patients and their families, who bear additional costs [17,18]. Cancer patients and their families assume certain additional expenses that, together with an irregular employment situation, can condition the evolution of the disease, even generating socio-economic risk or impact [19,20,21].

The Spanish Association Against Cancer [AECC], in its study on the financial toxicity of breast cancer, talks about the importance of quantifying the hidden suffering of this economic impact, highlighting how the patients in its study have seen their incomes decrease and their expenses increase. Household expenses (doctors, pharmacy, equipment, etc.) increase; in total, 92% of the patients in the study have expenses incurred in transportation, meals, and accommodation, 91% in ortho-prosthetic products or home adaptation and 29% in contracting healthcare personnel [16]. Wyman states how families of cancer patients bear up to 45% of the total cost of the disease in direct non-medical costs related to transportation, food, lodging, equipment, formal care, informal care, and transport to radiation therapy [1].

There are several unconsidered characteristics of cancer patients and their families that may affect the level of household income and the resulting additional expenditure, such as the level of education of the patient, the employment status of the patient and the level of family support in the evolutionary process of the disease. Therefore, the aim of our study was to analyse the variables level of education, employment status of the patient and main caregivers of the cancer patient and how these affect the level of household income, additional expenditure and socio-economic impact of the patient and family.

## 2. Materials and Methods

### 2.1. Design and Procedures

The research carried out is an epidemiological cross-sectional descriptive randomised observational study without replacement taking into account the anatomopathological diagnosis of cancer.

### 2.2. Participants

The study was carried out with patients admitted to the Medical Oncology Service, patients receiving active outpatient treatment at the Day Hospital, and patients receiving active treatment at the Radiotherapy Service at the University Hospital of Salamanca (CAUSA), Spain. Selection criteria:-Inclusion criteria: anatomopathological diagnosis of cancer, over 18 years of age, and signed voluntary consent to participate in this study.-Exclusion criteria: clinically impaired cognitive status (Mini-Mental State Examination (MMSE) score below 24 points), failure to correctly complete the assessment tools required for the study.-Withdrawal criteria: express request for withdrawal from the patient’s family, even if they had completed the informed consent document and/or did not correctly complete any of the assessment instruments required for this study.

### 2.3. Sample Size

The sample size was calculated based on the determination of the power of the study. The statistical parameters of confidence, and the probability of occurrence of the event, considering a maximum expected estimation error, were taken into account.

The size was according to the following considerations:-Total number of people with cancer in Spain between 1 January 2016 and 31 December 2020 was 867,656 people (Observatory of the Spanish Association Against Cancer, 2021) [15,21].-Total number of new cancer diagnoses in Spain was 285,658 people (Observatory of the Spanish Association Against Cancer, 2021) [15,21].-Total number of people diagnosed with cancer in Castilla y León between 1 January 2016 and 31 December 2020 was 54,049 people (Observatory of the Spanish Association Against Cancer, 2021) [15,21].-Total number of new cancers in Castilla y León is 17,888 people (Observatory of the Spanish Association Against Cancer, 2021) [15,21].-Total number of people with cancer in the province of Salamanca was 7043 people (Observatory of the Spanish Association Against Cancer, 2021) [15,21].

Based on these data, and following the formula for the calculation of the sample for a health study, which describes a descriptive qualitative study with a limited population:$$n=\frac{{N*z}_{\alpha^{2}}*p*q}{e^{2}*\left(N-1\right)+z_{\alpha^{2}}*p*q}$$

The factors that have been taken into account for the investigation and obtaining said sample size have been (Table 1): the number of people with a pathological diagnosis of cancer in Salamanca (N = 7043), estimating a confidence level of 95%, with a 50.00% probability that the event studied will result in success, a probability that the event studied (1-P) will not occur of 50.00%, and a maximum accepted estimation error of the 5.00%.

The different data applied in the formula determine that the optimal result of our study corresponds to a sample size of 365 patients:

We obtained the required sample size “n” = 364.33 persons.

### 2.4. Variables

#### 2.4.1. Study Variable

Additional financial costs resulting from a diagnosis of cancer disease for cancer patients and their families.

#### 2.4.2. Socio-Demographic Variables

Gender, age, marital status, main caregiver, level of education, level of dependency, the economic situation of patient and/or family, profession/employment status of patient and/or family, pensions/social security benefits, and health-related quality of life (HRQoL).

### 2.5. Measuring Instruments

To assess the different variables of the study, we used the following assessment and results collection tools:

Barthel Index (BI) [22]: used to measure a patient’s ability to perform basic activities of daily living (BADLs), taking into account whether he/she is fully independent, needs help or is dependent. The scale is divided into 10 items that correspond to the activities of daily living ABDVD evaluating for each of the patients: ability to bathe, cleanse, climb stairs, walk, move from bed to chair, use the toilet, dress, eat, and sphincter continence. The items are valued with a score of 0, 5, and 10 depending on whether it is partially independent or totally independent, respectively. The result of the summation can be: <20: degree of total dependency; 20–35 degree of severe dependence; 40–55: moderate degree of dependence; greater than or equal to 60: degree of slight dependence; 100: independent.

Lawton and Brody scale [23]: used to analyse the performance of the patient’s instrumental activities of daily living (IADLs). We have taken 8 items into account with this scale: taking care of the house, washing clothes, preparing food, going shopping, using the telephone, using transportation, handling money, and responsibility for taking medications. Patients have been assessed based on whether they can perform the task or not. The score can go from 0–1 points: dependent total, 2–3 points: severe dependence, 4–5 points: moderate dependence, 6–7 points: light dependence or 8 points: independence. We use this scale as a complement to the previous one to see the adaptation in the context of the cancer patient and the ability to maintain autonomy in a community environment.

ZARIT Primary caregiver overload Test [24]: used to assess primary caregiver burden for cancer patients. Interprets from 22 to 110 points representing objective caregiver burden. The maximum score can be 88: no overload, between 47–55 points: slight overload, and greater than 56 points: intense overload. We have used this measure for its reliability and validity, taking into account the Cornbach alpha coefficient of 0.91. In this way, we have been able to obtain information on the emotional part of the moment of evaluation and the weeks prior to this.

ECOG scale [12,25]: used exclusively for cancer patients to assess their general condition and quality of life. It takes into account capacities, paying attention to their autonomy. We have taken into account with this scale the objective part of assessing the patient’s quality of life, taking into account the patient’s functionality, ability to carry out self-care activities, presence of symptoms, or time spent in bed. The coding of the scale gives us a result from 0 to 5, being able to obtain Ecog 0: the patient is completely asymptomatic and capable of carrying out work and activities; Ecog 1: the patient presents symptoms that prevent him from carrying out complex tasks, although he performs daily activities normally; Ecog 2: the patient is unable to perform any work, has symptoms that force him to remain in bed for several hours a day, other than at night but not exceeding 50% of the day; Ecog 3: the patient needs to be in bed for more than half the day due to the presence of symptoms, he needs help for most of the activities of daily life; Ecog 4: the patient is 100% in bed, needing help for all activities of daily living; Ecog 5: deceased. The validity is high: a correlation of Kendall 0.75 with a high correlation with the Kamofsky Index.

EUROQOL-5D questionnaire [13,14]: used to measure health-related quality of life (HRQoL). The patient assesses his or her own state of health. The evaluation dimensions have been mobility, personal care, daily activities, pain/discomfort, and anxiety/depression. Each of the dimensions has been assessed by the level of severity noted as no problems, some problems, moderate problems, and serious problems. We have used this scale for simplicity because the time required to administer it is short, and it has the ability to measure physical, psychological, and social aspects without loss of response numbers.

Self-completed questionnaire: we obtain data through this form created for the study on additional cost, taking into account general data related to health and the economic situation of the household. The form had the following dimensions: patient identification data, main caregiver identification data, patient health data, employment situation, and economic situation of the patient and family. These dimensions, in turn, have items and sub-items that have been valued and evaluated objectively to obtain more information from the participants. The questionnaire has been the starting point for data collection to subsequently use measurement scales.

### 2.6. Procedure and Data Collection

The data collection was carried out in a single moment, and for a single occasion in a period that has allowed the data to be obtained, it has been in a single session. The patients and main caregivers who were selected because they met the inclusion and exclusion criteria were informed of the existence of the present study, as well as of the objective and voluntary nature of their participation in it.

The assessment and data collection consisted, firstly, of the voluntary signing of the informed consent form, then the completion of the self-completion questionnaire and, finally, of the evaluation through the different instruments used. This assessment was carried out in the following phases:-New patient admission to Medical Oncology. After analysing the adequacy of the newly admitted patient based on the inclusion and exclusion criteria, the research and main objective were presented to the patient, followed by the signing of the informed consent form and the passing of the self-completion questionnaire. Subsequently, results have been evaluated through the measurement scales (Barthel Index, Lawton–Brody, Zarit, ECOG, and EuroQol-5D).-Patients attending for active treatment at a day hospital (chemotherapy or radiotherapy treatment). Firstly, we proceed to analyse which patients meet the selection criteria for the study. Once it is confirmed that they meet the criteria, the study is explained to the patient so that he/she can sign the informed consent document. At this point, the evaluation of the different variables to be studied begins. Data from each of the study participants were collected and coded in the Microsoft Access database developed for the research. Statistical analysis was performed using IBM SPSS Statistics version 26.

### 2.7. Statistical Analysis

The tests used have been studied using the Shapiro–Wilk statistic to determine normality using parametric or non-parametric methods.

For Descriptive Statistics: With normally distributed variables, we used mean and standard deviation, and with non-normally distributed variables, median and interquartile range. Categorical variables are defined through frequencies and percentages.

For analytical statistics: If the results obtained were non-parametric (*p* < 0.05) with the correlation of two variables. Variables were recoded when the corresponding variable did not have sufficient numbers with the coherent capacity to evaluate.

Comparison of two or more means was performed using the *Mann–Whitney U-Test* and of three or more means using the Kruskal–Wallis test with equality of initial conditions when *p* > 0.05.

The comparison of two means was analysed: parametrically: with the T-Student statistic (both in repeated means and in independent groups), non-parametrically with the Mann–Whitney U statistic (independent groups) or the Wilcoxon T-test (repeated measures). The results obtained have been expressed with the value of the statistic with *p*-values and those data that are most interesting for the interpretation of the research.

## 3. Results

As shown in Table 2, the research had a sample of 365 patients in which 88 patients had digestive cancer (24.1%), 85 lungs (23.3%), 72 breasts (19.7%), 23 prostates (6, 3%), 11 central nervous system (3.0%), 34 haematological (9.3%), and 52 others (14.2%).

The level of education was 53.2% of patients with no or primary education, 25.8% of oncology patients with secondary education, and 21.1% of patients with higher education. The profession of the patients in the study revealed that 36.4% of patients were employed, 10.1% were self-employed, 53% were not in active employment, and 38.9% were experiencing other conditions. The employment status of the patients at the time of the study was 40.5% retired, 34% of patients with some kind of incapacity to work, 11.5% unemployed, 8.5% working, 0.5% working part-time, and 0.8% students.

The age of the patient’s main caregiver had a mean of 57.1 years, with caregivers up to 90 years old. The gender of the main caregiver of the sick patient was 33.4% male compared to a majority sample of 66.6% female main caregivers. Their marital status was 72% married, 14% single, 2.7% widowed, and 3% separated. The degree of kinship of these main caregivers was 86% first-degree, followed by 5.5% second-degree, and a minimum of 0.8% contracted caregivers.

A total of 274 patients reported spending on pharmacy and/or parapharmacy in the last year, with 87.6% spending up to EUR 300, 9.5% up to EUR 900, and 2.9% up to EUR 1500. A total of 171 patients reported spending on orthopaedic equipment in the last year up to EUR300 for 54.4%, up to EUR 900 for 34.5%, and up to EUR1500 for 11.1%. A total of 54 patients in the sample had extraordinary expenditure in the last year on home help with a third person, up to EUR 300 for 48.1%, up to EUR 900 for 24.1%, and up to EUR 1500 for 27.8%. A total of 236 patients in the sample have had expenditure on home adaptation or trips to the hospital, up to EUR 300 for 64%, up to EUR 900 for 18.6%, up to EUR 1500 for 6.8%, up to EUR 2400 for 4.2%, and up to EUR 4500 for 6.4%.

We contrast economic variables with respect to educational attainment. Normality indicates that the variable has three different levels of education (“No education/primary education”, “Secondary education”, and “Higher education”); we compare three or more means.

Table 3 indicates the degree of statistical significance of the contrast between the different economic variables and the study grouping variable. The important results are:

For the variable “Amount of annual net household income prior to diagnosis”, there is a lot of significance, and patients with a higher level of education have a higher level of household income. Among those with a higher level of education, those with “Higher education” have higher incomes (*p* < 0.05). For the variable “Amount of annual net household income in the last tax year”, there is a lot of significance; patients with a higher level of education have a higher level of household income. Among those with a higher level of education, those with “Higher education” have higher incomes (*p* < 0.05). For the variable “Extraordinary expenses in the last year on home help and/or patient accompaniment service”, patients with “secondary education” (31.00) have spent more in the last year on “home help and/or patient accompaniment service” than those who did not have any type of studies (20.71) (*p* = 0.017). Likewise, the comparison between patients with “higher education” (32.72) and patients with “no studies/primary studies” (19.87) (*p* < 0.05) was very significant, with those with more education having much more expenses. The comparison between those who had more studies determined that those who had “higher education” (102.80) spent more compared to “secondary studies” (70.10) (*p* < 0.05). For the variable “Extra expenditure in the last year on housing adaptations or hospital transfers” (*p* = 0.016), patients with a higher level of education always had a higher additional cost, except in the comparison between “no studies/primary studies” with “higher education” where there was no significance (*p* = 0.536). For the variable “Extraordinary expenditure in the last year on pharmacy and/or parapharmacy due to oncological disease” (*p* = 0.152) and for the variable “Extraordinary expenditure in the last year on orthopaedic material due to oncological disease” (*p* = 0.315) there was no significance with respect to the level of education.

The results obtained indicate that the higher the level of studies, the higher the economic income in the home both before having cancer and during the disease. In addition, the higher the level of studies, the greater the expense in “home help and/or patient accompaniment services” and the greater the expenses in “extra expenditure in the last year on housing adaptations or hospital transfers”.

Table 4 indicates the degree of statistical significance revealed by the contrast between economic variables and the patient’s profession:

For the variable “Amount of Net Annual Household Income Prior to Cancer Diagnosis”, we observed as those patients who were working on their own or on behalf of others had more income in the home before cancer than those who were in another type. There is a lot of significance when we compare “not in employment” (75.34) with “other” (106.46) (*p ≤* 0.05), understanding this last group as those patients who were retired, thus having more income than those who were not employed. We can summarize that they have more income than those who are working on behalf of others or their own, followed by those who are retired. For the variable “Amount of Annual Net Household Income in The Last Tax Year”, we observe that patients have higher incomes when they are working on behalf of an account or on their own; it was also significant whether the patient was in the retirement stage. It is very significant to compare the figures for “not in employment” (65.75) with “worker on behalf” (104.56) (*p* ≤ 0.05) or “not in employment” (70.77) with “other” (108.16) (*p* ≤ 0.05), with the understanding that the latter as a retirement stage. For the variables “Extraordinary expenditure in the last year on pharmacy and/or parapharmacy due to oncological disease” (*p* = 0.238), “Extraordinary expenditure in the last year on orthopaedic material due to oncological disease” (*p* = 0.846), “Extraordinary expenditure in the last year on home help and/or patient accompaniment service” (*p* = 0.687) and “Extraordinary expenditure in the last year on transfers to hospital” (*p* = 0.672) there is no significance.

The significance indicates that the level of household income was higher before cancer when the patient was self-employed or employed by someone else than when they were unemployed or in the “other” stage, considering the latter category as the retirement stage; more homes have been entered the study in which the patient before cancer was self-employed or worked for someone else. However, this has not led to higher expenses in any of the economic variables studied.

Table 5 indicates the degree of statistical significance revealed by the contrast between economic variables and the main caregiver:

For the variable “Amount of net annual household income prior to tax diagnosis” (*p* = 0.004), there is significance. The average range indicates that there is a higher amount of net annual household income prior to the diagnosis in the first-degree kinship caregiver degree (170.16) than in the second-degree (109.48).

For the variable “Amount of net annual household income during the last fiscal year” (*p* = 0.018), there is significance. The average range indicates that there is a higher amount of net annual household income during the last fiscal year in the first degree of kinship of the main caregiver (169.62) than in the second degree (117.80). There is also more income in the first degree (159.09) compared to the hired caregiver (44.50) (*p* = 0.022).

For the variable “Extraordinary expenditure in the last year on orthopaedic material” (*p* = 0.028), there is significance. The average range indicates that those primary caregivers who were married (77.00) spent more on orthopaedic material than those who were single (57.85). Those who were separated or divorced (22.92) also spent more than those who were single (12.93) (*p* = 0.003).

The significance indicates how those family units made up of a patient and a first-degree relative have higher household income and higher expenses on orthopaedic material.

Table 6 indicates that there is no statistical significance between economic variables with the measurement scales.

The measurement scales used to evaluate the patients have not turned out to be significant in our study; They have not been a factor to be considered as a possible additional expense for dealing with the cancer disease. The disability and dependency of the patients in the sample or has entailed an additional expense.

## 4. Discussion

The aim of our research was to analyse the following variables, educational level, employment status of the patient, and main caregivers of the cancer patient and how these affect the level of household income, additional expenditure, and socio-economic impact on the patient and family.

Research on oncological disease and the additional costs that it produces for the patient and the family begins to be important in the present century at the moment in which we start to consider the multiple fields that surround the sick patient and the different needs that arise in the evolution of their disease.

Cancer generates additional expenses not covered by the health care system related to pharmaceutical expenses, expenses in orthopaedic products and equipment, expenses in contracting personnel for patient care, and expenses in travel to the hospital; all of these additional expenses are assumed by the patient and the family [16,20,21]. The results obtained show how patients with higher education or secondary education have a higher level of income both before having suffered from the disease and once they are sick; this is an important fact that we can connect to their having greater purchasing power for any type of additional expense. Patients who have a higher education have better job opportunities in the market, jobs with a better ability to adapt if they get sick, and benefits for social security illness (disability) with better amounts for sick leave.

We found little research that specifically assesses the importance of educational level as a determinant of health that affects the patient’s economic status and ability to bear expenses. However, authors such as Rioja and co-workers [26,27,28], in a study on similar factors and Alzheimer’s disease, state that the level of education is directly related to socioeconomic status and the capacity to slow cognitive deterioration. Additionally, contributions by authors such as Atance and colleagues [29], also in relation to this disease, state that the family’s economic status is fundamental to assuming the costs derived from the condition, given that the family bears more than 60% of the direct global cost of the patient’s care.

Patients with a higher level of education in the sample had higher extraordinary expenses in the last year for home help and/or patient accompaniment services and also higher extraordinary expenses for transfers to the hospital, a fact that is related to having greater purchasing power and capacity to be able to make these expenses. These results coincide with the research of Salas and colleagues [30], who found that the quality of life during the oncological process was better for those patients with a higher level of education and a higher socioeconomic level to support their illness. It is also related to the research of Sharrocks et al. [31], who speak of the importance of the cancer patient’s economic capacity to be able to cover the costs of any clinical trial. We can affirm that the level of education and the financial capacity of the patient at home can determine the quality of life during the cancer disease process.

The employment situation at the time the patient receives the diagnosis is another one of the determinants that can condition the socioeconomic situation of the patient and the family, determining the evolution of the patient’s own disease [15,16,17].

The amount of income in the home of the cancer patients in the sample was higher both before having cancer and while they were sick, if the patient was working for someone else or self-employed compared to being unemployed or retired. However, having higher household income for the patients in the sample did not turn out to be related to having significant additional expenses at the pharmacy, expenses for in-home help and/or accompaniment service to the patient, expenses in transfers to the hospital, and other categories. This result is related to research by Ayala et al. [32] on the unmet needs of cancer patients in active treatment. They found that up to 46.95% of the patients evaluated reported that their needs were not met when the disease or its treatment imposed restrictions on physical activities, when financial resources were reduced, or when patients needed help from third parties other than family, all categories related to extraordinary expenses. In all, 31.79% of the patients in the sample were employed and/or self-employed; 54% of the patients stated that their physical needs were not met, and 38.14% stated that they needed support from a third person.

It is paradoxical to think that those patients with a higher level of education shown above, have higher extra expenses for home help and/or patient accompaniment services and also higher expenses for transfers to the hospital. Concurrently, having a higher household income has not been significant in terms of higher extra expenses in view of the needs of patients.

With regard to this last question, we consider the culture of informal family care of the sick patient; thus, our research had a total of 72.9% of caregivers in married marital status, 86% in the first degree of consanguinity, and 66.6% female primary caregivers. The study by Rodriguez et al. [33] affirms that the family is the basic pillar in the provision of care for cancer patients, with 82% of the sample in their study being women in a married marital status. Additionally, Valencia et al. [34] detail the relevant role of the spouse in this care, with up to 58% of the sample being married. Home help and/or accompaniment services for cancer patients, as well as transfers to the hospital, are directly related to the family as a basic pillar in the provision of care for cancer patients, which explains the low significance of the result.

Research has shown that those patients who had a primary caregiver in the first degree of consanguinity had higher household incomes, highlighting the important role of the family in the care of the sick patient. Those patients who had a primary caregiver in the first degree also had higher expenses in orthopaedic material, symbolising the family as a provision of this patient care service.

Ríos states that the patient’s family system, and specifically, the figure of the main caregiver, is paramount for the treatment and care of the sick patient, seeking better resources according to their needs and not only in extreme cases [35]. We did not find coincidence in terms of income and expenses according to marital status or consanguinity of the main caregiver, but, like most of the research to date, the patients are married, as in the study by Reina and collaborators [36] with up to 79.2% of patients married and/or in a consensual union or the study by Ayala with up to 62.42% of patients married and/or in a consensual union [32].

The care of the cancer patient needs to be considered from a holistic point of view, with the epicentre of care residing in the patient himself, taking into account multiple determinants that affect his health; Vicente et al., in a study regarding cancer patients [37], consider continuous care essential in each of its phases, considering all patient needs; Gallegos and collaborators [38] also affirm how there are gradually new orientations in the care of the sick patient leading doctors to consider the patient not as a mere carrier of a diseased organ, but as a person in its entirety who should be treated as such.

In short, patients with higher educational levels in the sample have been shown to possess higher household income and higher additional expenses for home help and/or patient follow-up and hospital transfer. Being self-employed or working for someone else led to a higher income but not higher additional spending on cancer needs. Married patients had higher household incomes. Based on the results obtained, we indicate that the level of studies, employment situation, income level, and the main caregiver are determinants that affect the disease and the quality of life of the cancer patient.

The limitations of the study have been the scarce bibliographical sources related to the subject and the target population of the study itself. Cancer is a complex disease to approach from any direction of research when the patient is ill. Future lines of research will focus on studying the family unit of the cancer patient based on the marital status of the sample and considering the level of education of the main carers and their income in the family unit. This is likely to influence the amount of income in the household, the additional expenses, and the way in which the evolutionary process of the disease is managed.

## 5. Conclusions

From the results obtained in our study, we can conclude that cancer patients in our sample with a higher level of education spent more on home help and/or a patient accompaniment service and also spent more on transfers to the hospital for sick patients to receive their treatment. In addition, in our sample, having a higher income at home for the cancer patient was not related to having higher expenses at the pharmacy, on orthopaedic materials, home help, and transfers to the hospital for treatment.

The main caregivers of the patient are a fundamental pillar in the evolution of the disease. Most of the caregivers in the sample were in the first degree of consanguinity, reflecting the Spanish culture of family care. Those patients who have had the family support of a primary caregiver in the first degree of kinship have had higher family income at home and higher spending on orthopaedic material.

Cancer generates economic expenses in the homes of oncology patients and their families who have to face, in addition to the evolutionary process of the disease, expenses related to pharmaceuticals, orthopaedic material, home help, and transfers to the hospital for treatment.

## Figures and Tables

**Table 1 healthcare-11-01306-t001:** Research parameters.

Parameter	Worth	Where	Investigation
N	7043	Population size	Number of people with an anatomopathological diagnosis of cancer in Salamanca in 2019.
Z	1960	Statistical parameter that depends on the Conficence Level (NC).	(Z-alpha): dependent on N, with a confidence level of 95%: 1.96%.
P	50.00%	Probability that the studied event occurs (success).	
Q	50.00%	(1-P) Probability that the event studied will not occur.	
e	5.00%	Maximun accepted estimation error.	

**Table 2 healthcare-11-01306-t002:** Socio-demographic variables of the study sample.

Socio-Demographic Variables	N	Results
Patient’s educational level	365	Without studies/primary studies: 53.2% Secondary studies: 25.8% Higher studies: 21.1%
Patient profession	365	Self-Employed labor active: 10.1% Employee employed: 36.4% Non-labor active: 53% Other: 38.9%
Current employment status of the patient	361	Active labor force: 8.5% Unemployed: 11.5% Short-time: 0.5% Student: 0.8% Disability: 34% Retirement: 40.5% Other: 3%
Gender of patient’s primary caregiver	338	Male 33.4% Female 66.6%
Marital status of patient’s primary caregiver	338	Singles: 14.0% Married: 72.9% Separated and divorced: 3% Widowed: 2.7%
Age of patient’s primary caregiver	337	Media = 57.19 years (±14,052) Max. = 90 years.
Degree of kinship primary caregiver of patient	337	Grade 1: 86%. Second degree: 5.5%. Contracted caregiver: 0.8%.
Amount of net household income during the last fiscal year	363	Less than EUR 12.000: 27.1% From EUR 12.001 to EUR 24.000: 47.9% More than EUR 24.001: 24.4%
Amount ot net household income before cancer diagnosis	363	Less than EUR 12.000: 21.9% From EUR 12.001 to EUR 24.000: 51.5% More than EUR 24.001: 26.1%
Extraordinary expenses in the last year in pharmacy and paraphamacy	274	EUR 300: 87.6% EUR 900: 9.5% EUR 1500: 2.9%
Extraordinary expenditure on orthopaedic equipment in the last year	171	EUR 300: 54.4% EUR 900: 34.5% EUR 1500: 11.1%
Extraordinary expenditure in the last year on home help and patient accompaniment service	54	EUR 300: 48.1% EUR 900: 24.1% EUR 1500: 27.8%
Average extraordinary expenditure on transfers to hospital in the last year	236	EUR 300: 64% EUR 900: 18.6% EUR 1500: 6.8% EUR 2400: 4.2% EUR 4500: 6.4%

**Table 3 healthcare-11-01306-t003:** Contrast statistics. Grouping variable: educational level.

Variable	Educational Level	N	Value *p*	Average Range
Amount of anual net household income prior to diagnosis	No primary education/studies	194	*p* = 0.003	133.66
	Secondary education (Baccalaureate/High School)	92		164.25

	No studies/primary studies	194	*p* = 0.000	114.05
	Higher education (University/Higher education)	77		191.29

	Secondary studies (Bach/Advanced)	92	*p* = 0.000	72.29
	Higher education (University/Higher)	77		100.19
Amount of annual net household income in the last tax year	No primary education/studies	194	*p* = 0.018	135.76
	Secondary education (Baccalaureate/High School)	92		159.83

	No studies/primary studies	194	*p* = 0.000	113.97
	Higher education (University/Higher education)	77		191.49

	Secondary studies (Bach/Advanced)	92	*p* = 0.000	70.10
	Higher education (University/Higher)	77		102.80
Extraordinary expenses in the last year on home help and/or patient accompaniment service	No primary education/studies	35	*p* = 0.017	20.71
	Secondary education (Baccalaureate/High School)	10		31.00

	No studies/primary studies	35	*p* = 0.002	19.87
	Higher education (University/Higher education)	9		32.72

	Secondary studies (Bach/Advanced)	10	*p* = 0.571	9.20
	Higher education (University/Higher)	9		10.89
Extraordinary expenditure in the last year on housing adaptations or hospital transfers	No primary education/studies	140	*p* = 0.016	96.05
	Secondary education (Baccalaureate/High School)	63		115.21

	No studies/primary studies	140	*p* = 0.336	88.45
	Higher education (University/Higher education)	33		80.85
	Secondary studies (Bach/Advanced)	63	*p* = 0.012	53.12
	Higher education (University/Higher)	33		39.68

**Table 4 healthcare-11-01306-t004:** Contrast statistics. Grouping variable: Patient’s profession.

Variable	Educational Level	N	Value *p*	Average Range
Amount of anual net household income prior to diagnosis	Own-account assets	35	*p* = 0.395	78.34
	Assets held for hire or reward	133		86.12

	Own-account labour assets	35	*p* = 0.003	53.91
	Not in employment	53		38.28

	Own-account labour assets	35	*p* = 0.547	93.61
	Other	142		87.86

	Employed persons in employment	133	*p* = 0.000	104.84
	Not in employment	53		65.04

	Employed persons in employment	133	*p* = 0.11	149.98
	Other	142		126.78

	Not in employment	53	*p* = 0.000	75.34
	Other	142		106.46
Amount of annual net household income in the last tax year	Own-account assets	35	*p* = 0.223	75.69
	Assets held for hire or reward	133		86.82

	Own-account labour assets	35	*p* = 0.015	52.37
	Not in employment	5335		39.30
		142		

	Own-account labour assets	133	*p* = 0.513	84.01
	Other	53		90.23

	Employed persons in employment	133	*p* = 0.000	104.56
	Not in employment	142		65.75

	Employed persons in employment	53	*p* = 0.196	144.26
	Other	142		132.13

	Not in employment		*p* = 0.000	70.77
	Other			108.16

**Table 5 healthcare-11-01306-t005:** Contrast statistics. Grouping variable: patient’s primary caregiver.

Variable	Degree of Relatedness of Patient’s Primary Caregiver	N	Value *p*	Average Range
Amount of annual net household income prior to diagnosis	Primary caregiver first degree of consanguinity	312	*p* = 0.004	170.16
	Primary caregiver second degree of consaguinity	20		109.48

	Primary caregiver first degree of consanguinity	312	*p* = 0.180	158.71
	Contracted caregiver	3		84.50

	Primary caregiver second degree of	20	*p* = 0.978	12.05
	Consaguinity Contracted caregiver	3		11.67
Amount of annual net household income in the last tax year	Primary caregiver first degree of consanguinity	312	*p* = 0.018	169.62
	Primary caregiver second degree of consaguinity	20		117.80

	Primary caregiver first degree of consanguinity	312	*p* = 0.022	159.09
	Contracted caregiver	3		44.50

	Primary caregiver second degree of	20	*p* = 0.187	12.75
	consaguinityContracted caregiver	3		7.00
Extraordinary expenditure on orthopaedic equipment in the last year	Single	23	*p* = 0.028	57.85
	Married	124		77.00

	Single	23	*p* = 0.003	12.93
	Separated and divorced	6		22.92

	Single	23	*p* = 0.614	14.65
	Widowed	6		16.33

	Married	124	*p* = 0.058	64.21
	Separated and divorced	6		92.25

	Married	124	*p* = 0.516	65.98
	Widowed	6		55.67

	Separated and divorced	6	*p* = 0.125	8.33
	Widowed	6		4.67

**Table 6 healthcare-11-01306-t006:** Contrast statistics. Grouping variable: measurement scales.

Variable	BARTHEL	N	Value *p*	LAWTON BRODY	N	Value *p*	ECOG	N		Value *p*	ZARIT	N	Value *p*
Amount of anual net household income prior to diagnosis	Total	48	*p* = 0.186	Total	32	*p* = 0.046	Ecog 0	59	*p* = 0.194		No overload	240	*p* = 0.029
	Severe	21		Severe	44		Ecog 1	125			Light overload	49	
	Moderate	47		Moderate	76		Ecog 3	89			Intense overload	74	
	Slight	131		Light	121		Ecog 4	48			Total	363	
	Independent	116		Independent	90		Ecog 5	2					
	Total	363		Total	363		Total	363					
Amount of annual net household income in the last tax year	Total	48	*p* = 0.475	Total	32	*p* = 0.081	Ecog 0	59	*p* = 0.424		No overload	240	*p* = 0.016
	Severe	21		Severe	44		Ecog 1	125			Light overload	49	
	Moderate	47		Moderate	76		Ecog 3	89			Intense overload	74	
	Slight	131		Light	121		Ecog 4	48			Total	363	
	Independent	116		Independent	90		Ecog 5	2					
	Total	363		Total	363		Total	363					
Change in revenue	Total	7	*p* = 0.540	Total	8	*p* = 0.273	Ecog 0	14	*p* = 0.152		No overload	74	*p* = 0.614
	Severe	10		Severe	11		Ecog 1	50			Light overload	17	
	Moderate	17		Moderate	32		Ecog 2	36			Intense overload	33	
	Slight	51		Light	48		Ecog 3	13			Total	124	
	Independent	39		Independent	25		Ecog 4	10					
	Total	124		Total	124		Ecog 5	1					
							Total	124					
Extraordinary expenditure in the last year pharmacy and parapharmacy	Total	36	*p* = 0.663	Total	27	*p* = 0.957	Ecog 0	35	*p* = 0.328		No overload	179	*p* = 0.095
	Severe	15		Severe	33		Ecog 1	99			Light overload	35	
	Moderate	43		Moderate	60		Ecog 2	67			Intense overload	60	
	Slight	102		Light	94		Ecog 3	36			Total	274	
	Independent	78		Independent	60		Ecog 4	35					
	Total	274		Total	274		Ecog 5	2					
							Total	274					
Extraordinary expenditure in the last year on orthopaedic equipment	Total	39	*p* = 0.451	Total	26	*p* = 0.142	Ecog 0	13	*p* = 0.693		No overload	92	*p* = 0.302
	Severe	11		Severe	29		Ecog 1	48			Light overload	32	
	Moderate	35		Moderate	43		Ecog 2	42			Intense overload	47	
	Slight	52		Light	43		Ecog 3	27			Total	171	
	Independent	34		Independent	31		Ecog 4	39					
	Total	171		Total	171		Ecog 5	2					
							Total	171					
Extraordinary expenditure in the last year on home help and patient accompaniment service	Total	11	*p* = 0.766	Total	7	*p* = 0.955	Ecog 1	12	*p* = 0.090		No overload	30	*p* = 0.621
	Severe	4		Severe	8		Ecog 2	21			Light overload	7	
	Moderate	14		Moderate	23		Ecog 3	10			Intense overload	17	
	Slight	20		Light	15		Ecog 4	11			Total	54	
	Independent	5		Independent	1		Total	54					
	Total	54		Total	54								
Extraordinary expenditure on transfers to hospital	Total	37	*p* = 0.885	Total	26	*p* = 0.850	Ecog 0	32	*p* = 0.024		No overload	146	*p* = 0.806
	Severe	16		Severe	34		Ecog 1	72			Light overload	33	
	Moderate	31		Moderate	48		Ecog 2	65			Intense overload	57	
	Slight	92		Light	81		Ecog 3	26			Total	236	
	Independent	60		Independent	47		Ecog 4	39					
	Total	236		Total	236		Ecog 5	2					
							Total	236					

## Data Availability

The data presented in this study are available on reasonable request from the corresponding author. The data are not publicly available due to the applicable data protection law.

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
