# Peer review of "Multifactorial Study on the Impact of Educational Level, Employment Status, and the Need for Extraordinary Care on the Economic Impact of Cancer Patients"

_healthcare, 2023, doi:10.3390/healthcare11091306_

Round 1

Reviewer 1 Report

The manuscript made by García-Martín A et al., is interesting since they are writing about the association between several economic and social variables with cancer. Despite the interest of the manuscript, the authors need to resolve some suggestions.

First.

2.3 Sample Size

The authors explain the relationship between the cancers of Spain (Castilla y León) and Salamanca, but for what purpose do they explain these?

When the authors describe five years, I recommend using specific data, instead of five years, for example:

"Total number of people..." Instead, use the last five years: January 1st, 2023, to December 31st, 2028.

Second.

"We have taken..." This paragraph is not comprensible. I suggest redacting better and establishing a relationship between this paragraph and point 2.3.

Third. How determined the sample size? If the formula was used, please explain better, for example:

n= Nzα2 pq 104 e2 (N−1)+zα2 pq. Please use numbers in this formula and explain in detail how the sample size was calculated. 

Fourth.

3. Results: “52 others (14.2%)…” Would it be possible to explain the other tumors, or at least the three most commonly occurring tumors? 

Fifth.

5. Discussion.

"Sha-rrocks..." Please review this citation and use et al., instead of collaborators.

Overall comments

The study made by Garcia-Martin et al., as mentioned in a comment above, is interesting; however, this manuscript has several mistakes that the authors need to review.

This is a descriptive study, and the statistical study is not adequately described; it is strongly recommended that it be included in Table 2 as the type of most common cancer, with the variable in bold, for example:

"Amount of annual net household income," cancer most frequently reported, education level, etc.

The manuscript needs to be reviewed by an expert in the language; it has grammar mistakes.

Author Response

POINT BY POINT RESPONSE TO REVIEWER

REVIEWER 1:

The manuscript made by García-Martín A et al., is interesting since they are writing about the association between several economic and social variables with cancer. Despite the interest of the manuscript, the authors need to resolve some suggestions.

#1: 2.3 Sample Size. The authors explain the relationship between the cancers of Spain (Castilla y León) and Salamanca, but for what purpose do they explain these?

When the authors describe five years, I recommend using specific data, instead of five years, for example: "Total number of people..." Instead, use the last five years: January 1st, 2023, to December 31st, 2028.

Response: Thank you very much for the suggestion. We will make the changes as you have indicated. Salamanca is one of the provinces of Castilla y León. We have decided to consider the number of patients from the Autonomous Community of Castilla y León and the province of Salamanca in order to obtain a sufficient number of patients for the study through the formula.

#2: "We have taken..." This paragraph is not comprensible. I suggest redacting better and establishing a relationship between this paragraph and point 2.3.

Response: Thank you very much for the suggestion. We have made the suggestions so that the paragraph can be better understood.

#3: How determined the sample size? If the formula was used, please explain better, for example: n= N∗zα2 ∗p∗q 104 e2 ∗(N−1)+zα2 ∗p∗q. Please use numbers in this formula and explain in detail how the sample size was calculated. 

Response: Thank you very much for the suggestion. We have completed requested information.

#4: 3. Results: “52 others (14.2%)…” Would it be possible to explain the other tumors, or at least the three most commonly occurring tumors? 

Response: Thank you very much for the input. We have made the difference between the main diagnoses and we have considered it appropriate not to make a greater difference with those of lesser importance. However, we will consider it for the following investigations.

#5: 5. Discussion. "Sha-rrocks..." Please review this citation and use et al., instead of collaborators.

Response: Thank you very much for the input. We make the commented modification.

Overall comments

The study made by Garcia-Martin et al., as mentioned in a comment above, is interesting; however, this manuscript has several mistakes that the authors need to review. This is a descriptive study, and the statistical study is not adequately described; it is strongly recommended that it be included in Table 2 as the type of most common cancer, with the variable in bold, for example: "Amount of annual net household income," cancer most frequently reported, education level, etc. The manuscript needs to be reviewed by an expert in the language; it has grammar mistakes.

Response Thanks a lot for the suggestion. We have proceeded to make clarifications about the commented author and to rewrite literature to be more specific.

Reviewer 2 Report

The paper considers the important supporting role  of the family in the patient's care affected by cancer. This aspect appears to be neglected by literature. Then the article could stimulate future researches and this is useful to meliorate the supporting strategy also in the families that cannot do. 

Please you should correct in the table 1 Age of patient's primary caregiver "anos" with years

line 204 the sentence is not clear perhaps you should eliminate the dot, I don't understand.

Author Response

POINT BY POINT RESPONSE TO REVIEWER 2

REVIEWER 2:

The paper considers the important supporting role of the family in the patient's care affected by cancer. This aspect appears to be neglected by literature. Then the article could stimulate future researches and this is useful to meliorate the supporting strategy also in the families that cannot do. 

#1: Please you should correct in the table 1 Age of patient's primary caregiver "anos" with years.

Response: Thank you very much for the suggestion. We have proceeded to make commented changes.

#2: line 204 the sentence is not clear perhaps you should eliminate the dot, I don't understand.

 Response: Thank you very much for the suggestion. We have proceeded to make commented changes.

Round 2

Reviewer 1 Report

The authors responded adequately to all my questions and suggestions; the manuscript has improved and is now better redacted.